# Developing an Online Dashboard to Visualize Performance Data—Tennessee Newborn Screening Experience

**DOI:** 10.3390/ijns8030049

**Published:** 2022-09-02

**Authors:** Charles Lechner, Marc Rumpler, Mary Christine Dorley, Yinmei Li, Amanda Ingram, Hilary Fryman

**Affiliations:** Tennessee Department of Health, Nashville, TN 37216, USA

**Keywords:** data visualization, quality improvement, newborn screening, tableau

## Abstract

Newborn screening (NBS) is a vital public health program and delays in the screening process can lead to catastrophic outcomes for infants and their families. Efforts to improve screening quality in Tennessee are proactive and ongoing. From these efforts, an open-access dashboard has been developed to address a need for methods to better visualize performance data, promote data transparency, and drive quality improvement. Dashboard development was a collaboration between a fellow from the Association of Public Health Laboratories (APHL) and Tennessee NBS staff. Iterative dashboard prototypes were developed using Tableau software and incorporated feedback from Tennessee birthing facility staff and health experts. Infrastructure and procedures were created to reduce the burden of future dashboards. Eight NBS performance indicators are visualized across several views. These views are designed to provide an overview of NBS performance data when first accessed, then allow for a drill-down into specific data. This dashboard drives introspection at the state and facility level, making it possible to identify potential issues and necessary corrective actions earlier, therefore improving the completeness and timeliness of NBS in Tennessee. The experiences from developing this dashboard can be applied to future dashboard development in Tennessee NBS and other public health programs implementing similar measures.

## 1. Introduction

Newborn screening (NBS) is the process of identifying infants at risk for certain genetic disorders, assisting with clinical diagnoses of such disorders, and providing treatment or dietary interventions to limit or prevent harmful outcomes. NBS is a critical public health program. Every year in the US millions of newborns are screened, and thousands of infants’ lives are saved and improved, as a result [1,2]. Effective operations and continuous improvements of public health programs, such as NBS, require the availability of detailed program performance data [3]. Due to the population-wide, surveillance-based nature of NBS, such data are readily available even at the level of an individual state. The Tennessee NBS program screens roughly 85,000 newborns each year and when including repeat samples, there are nearly 90,000 infant specimens from which data are generated. These specimens come from sixty-five facilities to a central NBS Laboratory where they are screened for seventy-one conditions. The rapid accumulation of such vast amounts of data poses challenges as to how to readily extract relevant information so that it can be effectively used by program decision-makers [4,5]. The deployment of visualization tools and intelligent use of programmatic data are already widely employed in public health and could be adapted to improve NBS outcomes.

Data visualization can be defined as the use of visuals to aid in task completion through cognitive amplification [6]. By converting data into information through data visualization, vital information can be promptly disseminated and viewed by relevant stakeholders. Visualization tools take data and create interactive visual representations of these data. In turn, these created visuals take advantage of human spatial and visual cognitive processes to build mental models of the data in the user’s mind that aid in decision-making [7,8,9]. Research has shown that information presented in a graphical format can be processed more readily by humans than if it was presented in the text [10,11]. When it comes to large complex datasets, data visualization plays a key role in finding new insights [12]. Datasets used by NBS programs fall into this category. They consist of data from state screening laboratories, follow-up programs, individual facilities within states, and state birthing records for thousands of births each year. A data dashboard is an ideal tool when visualizing these datasets as it allows for the consolidation of data from multiple sources into a single display that promotes understanding and monitoring of the whole data system [13,14,15].

In some aspects of NBS performance, quality improvement (QI) measures are needed and tools that can assist with these measures are highly desired. On the status of screening timeliness in United States NBS programs, a Government Accountability Office report from 2016, and the 2019 and 2020 NewSTEPs Annual Report, have shown that many state NBS programs are struggling to meet goals set by the Advisory Committee on Heritable Disorders in Newborns and Children (ACHDNC) [16,17]. While these goals are not mandatory across the United States, they are recommendations that are expected to be strictly adhered to to ensure infants are screened and results are reported by an infant’s seventh day of life. Failure to meet these goals could negatively impact infant health and even result in death [18]. Data dashboards can be an effective solution to address this critical issue as they are data-driven, evidence-based tools that assist with effective decision-making, thereby leading to improved NBS quality and positive impacts on infants’ lives.

This paper details the efforts and experiences of Tennessee NBS in developing and deploying a dashboard to visualize NBS performance data. This dashboard was released in May 2022 as an open-access tool to drive QI and better decision-making within the program. While the dashboard has already made an impact at the state level, it is intended to be a tool for individual facilities in Tennessee. It provides a novel, interactive, and simple way for facilities to monitor their performance and identify areas of NBS for which improvement is needed. Despite individual facilities being the primary audience for the dashboard, it is available online to the public to promote the transparency of NBS performance. The dashboard provides a unique peek into the performance of a state health program and allows Tennessee parents to make impactful perinatal care decisions if given the opportunity to choose a Tennessee birthing facility. By documenting the dashboard’s development in this paper, we hope that it can be used as a learning experience for other public health programs seeking to implement similar projects.

## 2. Materials and Methods

When developing a dashboard, several aspects must be considered: the design of the underlying data tables, the level of user interaction, dashboard scope and accessibility, existing state and departmental procedures, regulations, and contracts, technological infrastructure requirements and restrictions, and more, all contribute to the development and success of a dashboard. Tennessee NBS divided this project into three stages to tackle these developmental complexities.

Dashboard planning.Dashboard building.Dashboard publishing.

A process map of the major steps taken during dashboard development is shown in Figure 1.

### 2.1. Dashboard Planning

This stage took one month to complete. At this stage, many of the decisions contributing to the success of the dashboard, such as what metrics to include, how to manage the dashboard’s data, and how to design the dashboard, are discussed and documented in a project plan. This plan remained flexible so that any changes made during development could be incorporated easily. The personnel and tools necessary for dashboard development are also defined and acquired.

To build an effective dashboard, experience with data visualization techniques and software, as well as profound knowledge of NBS Laboratory and follow-up program data and processes, is highly desirable. A dashboard workgroup was assembled consisting of a fellow from the Association of Public Health Laboratories (APHL) NBS Bioinformatics and Data Analytics Fellowship, key leadership from the Tennessee NBS Laboratory and follow-up programs, and an epidemiologist and nurse education from the Follow-Up Program. The software to develop the dashboard was limited by what was accessible through Tennessee state procedures and contracts. For data processing and management, the analytics software SAS (v9.4, produced by SAS Institute, Cary, NC, USA) was available and widely used in the Tennessee state government. For building the dashboard, the data visualization software Tableau (v2021.2, produced by Salesforce, Seattle, WA, USA) was used as it has gained increasing popularity and technical support.

Past public and internal reports were examined to identify key metrics to include in the dashboard [19]. Metrics were chosen based on their ability to provide an overview of NBS program performance and how targetable, at the state laboratory or individual facility level, the metric could be for quality improvement efforts. The data tables used for the dashboard were derived by linking Tennessee birth records to internal NBS records containing screening results and specimen handling information for each infant. These tables are updated monthly. To preserve data confidentiality and consistency with prior Tennessee NBS reports, the lowest level of data summarization within the dashboard would be monthly, though quarterly and yearly summarization is also available. The visual design principle of providing an overview first, followed by the ability to zoom and filter, and then providing additional details on demand was used when designing the dashboard to limit data overload [20]. This principle was applied to each view in the dashboard and the dashboard’s overall layout. When designing the dashboard, existing dashboards were examined to draw inspiration for the NBS dashboard. These included dashboards on COVID-19 from within the Tennessee Department of Health and other public healthcare providers, such as Johns Hopkins University, NewSTEPs dashboards on NBS programs in the United States, and various dashboards available through the Tableau Public Viz Gallery [21,22,23].

### 2.2. Dashboard Building

This stage took six months to complete. Prototypes of the dashboard were developed iteratively and analyzed to determine if they met project goals. Necessary changes were then discussed for implementation in a future prototype. The APHL fellow was responsible for prototype building and the dashboard workgroup met weekly to analyze and discuss prototype development. To focus dashboard building efforts, features within individual dashboard views were built one at a time until they were deemed complete. During dashboard workgroup discussions, the entirety of the dashboard was regularly reviewed to assess necessary changes to any features previously deemed complete.

During dashboard building, feedback on features and layout was gathered via a survey (see Appendix A Table A1) sent to 68 NBS staff at Tennessee facilities. Dashboard progress was presented to a subset group of 30 staff from the original survey. Additional feedback was obtained by presenting the dashboard to the Tennessee Genetic Advisory Committee (GAC), a group of health experts who advise on Tennessee NBS. All input was incorporated into the dashboard building.

### 2.3. Dashboard Publishing

After the dashboard was finalized on a local system, the development moved to publishing. This stage focused on ensuring external systems, procedures, and requirements were met so that the dashboard could be published. Many aspects of this stage involved collaborating with other workgroups in the Tennessee state government. This stage took two months to complete.

With the help of Tennessee’s IT department, a project space was reserved on a state Tableau test server available through the Tennessee Department of Health. Data tables were moved to network storage drives and appropriate connections were made to ensure the Tableau server had access. The dashboard was uploaded to the server project space and verified as working as intended. Data table updates on the network drive were scheduled to occur monthly. The Tableau server was scheduled to check the data tables nightly for any changes. This schedule was chosen as the best fit from a list of predefined schedules.

From the Tableau server, the dashboard was shared with director-level leadership so it could be approved for publishing. At the same time, approval was sought from the state’s branding workgroup. While waiting for approvals, the dashboard workgroup developed and disseminated resources to promote the dashboard’s release.

After approvals were acquired, the dashboard workgroup collaborated with IT to build a web page to host the dashboard. Once the web page was finalized, the dashboard was migrated to a production Tableau server and embedded in the web page. A dashboard user feedback survey was placed on the web page alongside the dashboard to gather feedback for future improvement.

## 3. Results

After nine months of planning and development, the dashboard was released online on the Tennessee Department of Health Newborn Screening Program website at the beginning of May 2022 [24].

### 3.1. Dashboard Indicators

Eight indicators are visualized in this dashboard: number of births, dried blood spot (DBS) screening rate, number of DBS, unsatisfactory specimen rate, initial specimens collected within the target time frame, all specimens received within the target time frame, all results reported by day of life (DOL) 7, and time-critical, presumed positive results reported by DOL 5. Table 1 describes these indicators further. When applicable, indicator goals have been set based on recommendations by the ACHDNC and Tennessee GAC [18].

### 3.2. Dashboard Views

#### 3.2.1. Main Page

The main page of the dashboard (see Figure 2) provides a snapshot view of the recent NBS performance. The most recent three months of data for all eight indicators summarized monthly are visualized on this page. Statewide data are shown by default. Users can visualize data for individual Tennessee facilities by selecting a facility of interest using a drop-down box. If there are no data for an indicator, the text will replace the indicator explaining why. For the DOL 5 indicator, only statewide data are available to preserve patient confidentiality and statistical reliability since this indicator is based on a small number of abnormal results. Indicators with state goals have color shading and conditional formatting applied to the data points to make it clear when a goal has been met. From the main page, users can navigate to the other views within the dashboard using buttons.

#### 3.2.2. Individual Indicator Views

Each indicator in the dashboard has its own view accessible from the main page. Figure 3 shows an example of one of these views. Data going back to January 2015 are available for these views. By default, the most recent twelve months of data for the state are displayed by month. Sliders allow for the date range to be adjusted and a drop-down box can be used to change the data summarization from monthly to quarterly or yearly. Individual facility data can be visualized by selecting facilities of interest from a checkbox list. This list can be filtered by facility size, determined by the number of births a facility experienced in the past year, and facility location. Data for individual facilities are represented by one line per facility by default. Users have the option to aggregate the data into a single line for all selected facilities with a drop-down box. The designs of the visualization for individual facility data were influenced by the survey sent to Tennessee facility NBS staff (see Appendix A Table A1). Features allowing for facility filtering and comparisons were regarded favorably in the survey and were prioritized when building this view.

#### 3.2.3. Facility Information Table

The facility information table displays contextual information for Tennessee facilities (see Figure 4). Facility name, location, and size are described with a table and a map. The specific number of births that determine a facility’s size is also defined in this view.

#### 3.2.4. Facility Rankings

The facility rankings view orders Tennessee facilities based on a user-selected indicator (see Figure 5). This ordering is visualized in a table and bar chart with additional contextual information shown based on the selected indicator. Facilities can be ranked on the DBS screening rate, unsatisfactory specimen rate, initial specimens collected within the target time frame, and all specimens received within the target time frame. By default, facilities are ranked by the unsatisfactory specimen rate performance from the previous twelve months. Users can change the time frame for ranking, in monthly increments, using sliders.

#### 3.2.5. Report Generator

The report generator view displays all eight indicators in a vertical format suitable for exporting to PDF. By default, the indicators show the past twelve months’ worth of data for the state by month. Users can change the time frame being displayed, the data summarization from monthly to quarterly or yearly, and the facility being visualized on this view with drop-down boxes and sliders.

## 4. Discussion

Within two weeks of being published, the dashboard has made three significant impacts on Tennessee NBS. First, a previously unnoticed error in data processing methods when linking birth records to NBS records was found and corrected. Second, NBS staff utilized the dashboard to generate visuals of program performance for reports given during internal meetings. This saved a significant amount of time compared to older methods used as self-reported by the staff. Third, the dashboard was used to identify facilities that have continually struggled with specimen transit timeliness without having to manually assemble reports from individual facilities. This allowed for NBS education efforts to be tailored to the facilities in need of them. Such early successes speak to the utility provided by the dashboard.

NBS QI using this public dashboard will continue. Procedures for monitoring state and facility performance and identifying corrective actions needed based on trends observed in the dashboard are being developed to assist with future QI efforts. The dashboard user feedback survey hosted online with the dashboard, along with a planned survey to Tennessee facilities to identify impacts and facility experiences, will be used to update the dashboard so that it can better fit the needs of our users.

Throughout the dashboard’s development, Tennessee NBS has documented aspects of the development process that we believe contributed to the dashboard’s success. First, the importance of researching existing dashboards when initially designing the dashboard. This allowed the workgroup to better understand what dashboards currently in public use look like, how effective they are, and what functionality is possible with Tableau. Second, is the importance of understanding departmental and jurisdictional regulations when it comes to developing an open-access tool that contains sensitive information. During dashboard planning, relevant IT and leadership workgroups were contacted, and documentation was sought, to define the regulations for this project. It was a lesson learned that verbal or email correspondence should be used to confirm regulations that are unclear or not fully defined in written documentation. During dashboard planning, documented restrictions on the abilities of the Tableau production server were not fully understood but were encountered during dashboard building. This resulted in the reworking of some dashboard features, which could have been avoided if restrictions were verified during planning. Third, is the importance of the dashboard workgroup’s composition and meeting schedule. By including leadership from the NBS Laboratory and follow-up program, as well as the nurse educator, a clear understanding of the dashboard’s purpose and requirements could be defined. A weekly meeting schedule for the workgroup allowed for constant development feedback and frequent collaboration. Finally, the importance of gathering input from the dashboard’s intended audience, Tennessee facilities. The dashboard development feedback survey and focus group presentations provided insight into future users’ needs and wants for the dashboard. Positive feedback on features affirmed the workgroup’s initial planning and questions or concerns identified areas of the dashboard that needed more thought.

Several limitations and circumstances unique to Tennessee NBS were encountered during the dashboard’s development. Other public health programs seeking to adopt similar measures may find them applicable to their programs. First, thanks to the availability of an APHL fellow and the preexisting infrastructure of a Tableau server, the cost of developing this dashboard was significantly reduced. Most costs were attributed to obtaining software licenses for the fellow’s local workstation. Second, for the dashboard to run efficiently, all Tennessee NBS data could not be included. Only data from 2015 onward was included to ensure performance would remain quick even after future data updates. In addition to this, to keep the dashboard efficient and prevent information overload for users, the level of detail for dashboard indicators had to be limited. Priority was given to aspects of the data that Tennessee facilities would find most useful. Third, data for time-critical results reported by DOL5 for individual facilities could not be included as there are very few of these results at each facility. To preserve patient privacy and to prevent reporting of data that would not have statistical use, it was decided to only report this indicator at the state level. Finally, the decision as to which indicators to include on the dashboard also took into account the preservation of patient privacy. NewSTEPs defines eight quality indicators to monitor and compare NBS program performance [27]. However, including data on infant diagnoses and follow-up care or on specimens that were flagged during the screening was not deemed appropriate by Tennessee NBS leadership for a dashboard that would be available to the general public, so only a few of the recommended indicators by NewSTEPs are included.

Since the processes for developing a data visualization dashboard in Tennessee NBS have been experienced, future dashboards are planned and under development. These dashboards are intended to be internal, containing sensitive data with levels of detail that would not be appropriate for a dashboard used by Tennessee facilities or the public. Such dashboards will allow for better insights into program performance and help to further drive NBS quality.

## 5. Conclusions

In this article, we outlined the experiences and processes undertaken by Tennessee NBS as we developed a data visualization dashboard for NBS performance data. This dashboard is a tool to help drive better decision-making and QI efforts at the state and individual facility levels. The dashboard has had an immediate impact at the state level, resulting in improved NBS quality and care for Tennessee infants. As an open-source data dashboard for NBS performance data, this tool appears to be novel among state NBS programs. The processes and experiences from developing this dashboard are already being used at Tennessee NBS to develop new tools to improve program quality. We hope they can be used by other public health programs to further their own public health goals and initiatives.

## Figures and Tables

**Figure 1 IJNS-08-00049-f001:**
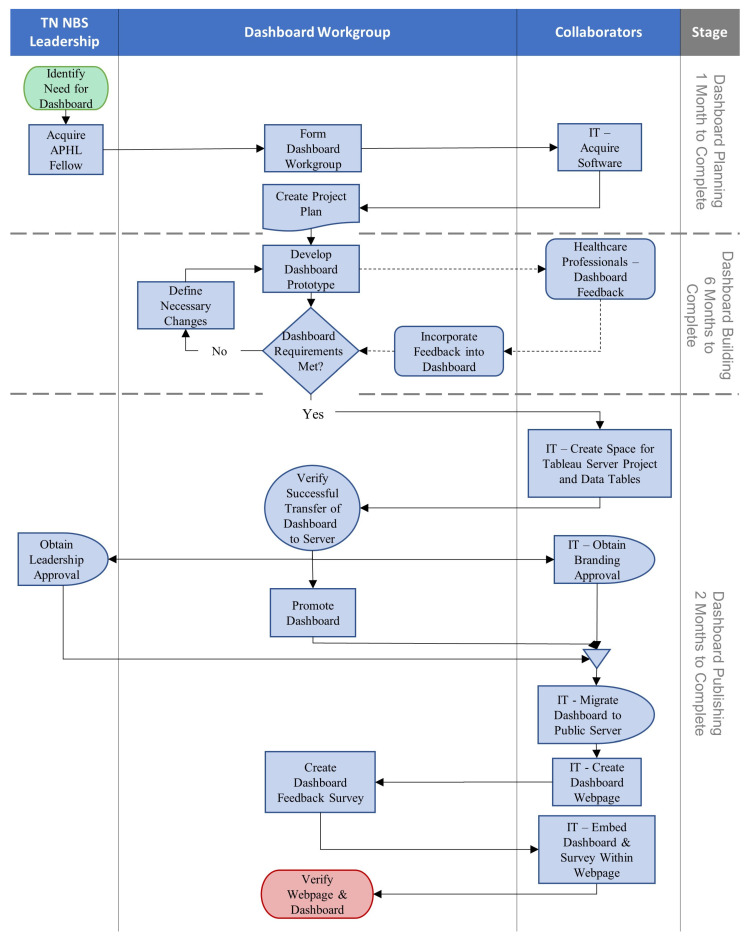
Tennessee (TN), Newborn Screening (NBS), Association of Public Health Laboratories (APHL), Information Technology (IT). A process map documenting the major steps taken during the dashboard’s development. Steps are grouped based on the dashboard development timeline and who was responsible for step completion.

**Figure 2 IJNS-08-00049-f002:**
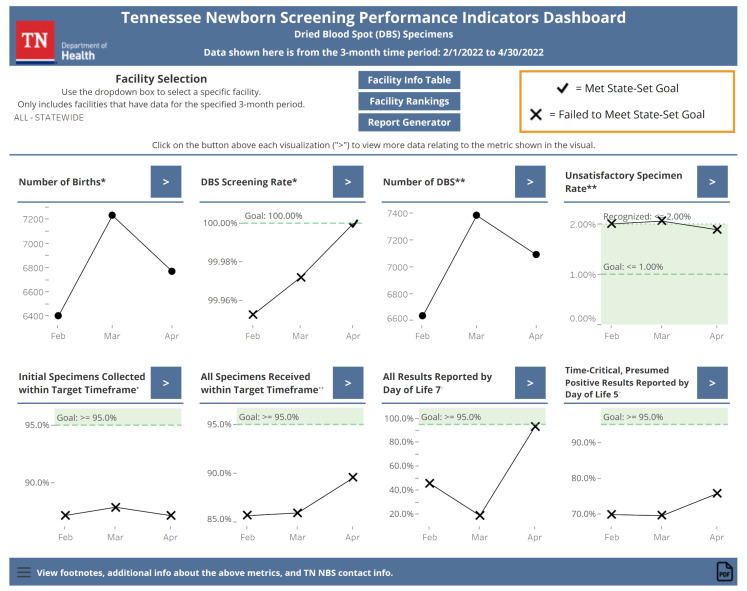
The default view of the dashboard’s main page as of June 2022. The three most recent months’ worth of data for all eight indicators are shown here with conditional formatting to indicate goal status where applicable. Buttons allow the user to navigate to other dashboard views. Markings like “*” and “**” indicate that the dashboard has footnotes that contain contextual information, similar to that given in Table 1, about the marked indicator.

**Figure 3 IJNS-08-00049-f003:**
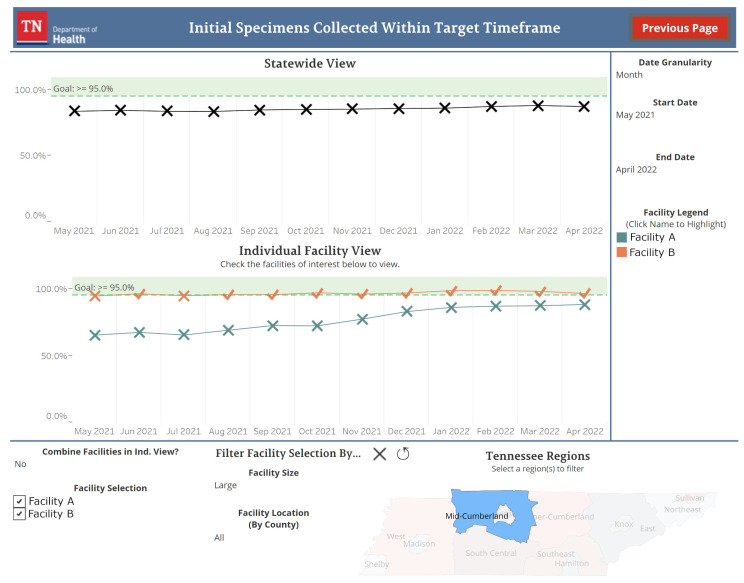
An example of the initial specimens collected within the target time frame indicator view. Individual facilities have been added after filtering for large-sized facilities in the Mid-Cumberland Tennessee region. Facility names in this figure have been anonymized for use in this manuscript. The time period and date granularity have not been adjusted from their default values.

**Figure 4 IJNS-08-00049-f004:**
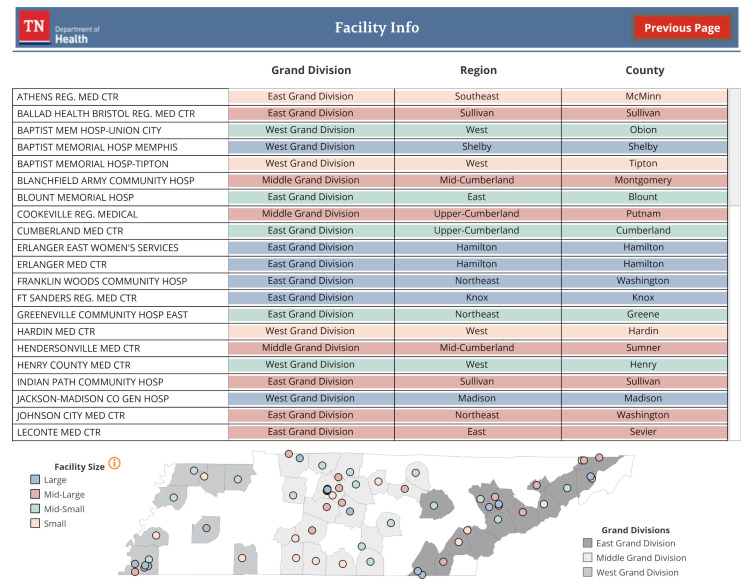
Information on Tennessee facilities’ county, region, and grand division location is available in a table and map. Facility size is indicated by table row and map dot color.

**Figure 5 IJNS-08-00049-f005:**
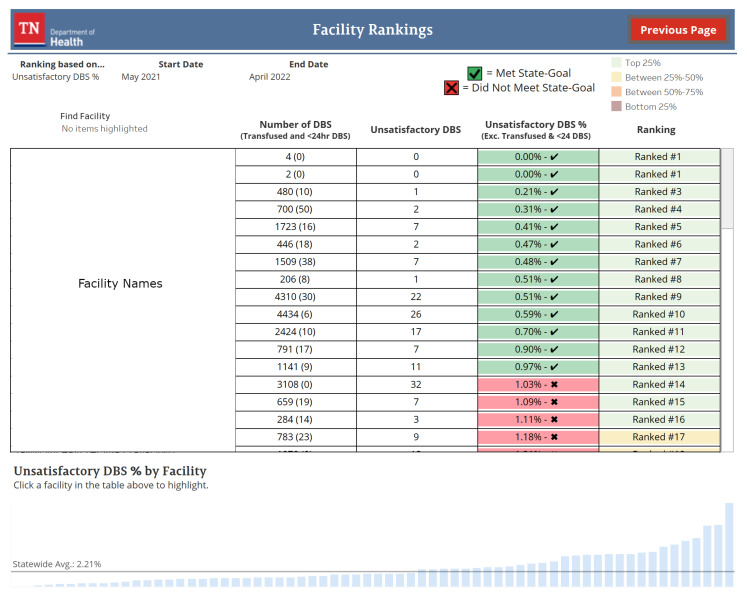
Dried blood spot (DBS). The default view of the dashboard’s facility rankings as of June 2022. Facilities are ranked by a user-selected indicator and displayed in a table and bar chart. Relevant information for the selected indicator is also displayed. Facility names in this figure have been anonymized for use in this manuscript.

**Table 1 IJNS-08-00049-t001:** Dried blood spot (DBS), day of life (DOL). The eight indicators available in the dashboard are described here. Indicator-specific details are explained in the description when applicable. Each indicator’s datum is grouped based on either the facility in which an infant was born, where the infant’s DBS was collected, or where the infant’s DBS was submitted from.

Indicator	Indicator Description	Indicator Grouped by…	State Goal
Number of births	Number of infants born.	Birthing facility	N/A
DBS screening rate	Percentage of infants who received a DBS screening. Does not include infants who expired before screening or whose parents refused screening. ^1^	Birthing facility	100%
Number of DBS	Number of DBS collected.	Collection facility	N/A
Unsatisfactory specimen rate	Percentage of DBS classified as unsatisfactory by state NBS Laboratory staff. ^2^ Does not include DBS collected less than 24 h after an infant’s birth or who received a blood transfusion as they produce unreliable results [25]. DBS collection is mandated even if a transfusion occurs and before an infant is transferred to another facility resulting in these DBS.	Collection facility	1% or less
Initial specimens collected within the target time frame	Percentage of DBS collected within 24–36 h of an infant’s birth. Only includes the initial DBS.	Collection facility	95% or more
All specimens received within the target time frame	Percentage of DBS received by the state lab within 48 h of collection. Includes initial and repeat DBS.	Collection facility	95% or more
All results reported by DOL7	Percentage of all results reported by an infant’s 7th DOL.	Submitting facility	95% or more
Time-critical, presumed positive results reported by DOL5	Percentage of results for specimens flagged positive for time-critical disorders, as defined by NewSTEPs, reported by an infant’s 5th DOL [26]. Only state-level data are included due to the small number of these results at individual facilities.	Submitting facility	95% or more

^1^ These infants are not included when calculating the DBS screening rate so as to not hold facilities accountable for circumstances out of their control. However, the exact numbers for these infants are available in the form of tooltips for this indicator. ^2^ Unsatisfactory specimens include blood spots whose quality is too poor for screening (e.g., contaminated, clotted, exhibits separated serum, or is otherwise altered or non-uniform), whose form information is incomplete or inaccurate, or whose specimens are mishandled (e.g., packaged improperly). Exact numbers for each unsatisfactory category are available in the form of tooltips for this indicator.

## Data Availability

Not applicable.

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
