# Peer review of "Developing an Online Dashboard to Visualize Performance Data—Tennessee Newborn Screening Experience"

_2409-515X, 2022, doi:10.3390/ijns8030049_

Round 1

Reviewer 1 Report

Dear Authors:

I was very excited to see this article pop up in my in-box. I think this is important work and will help the newborn screening world so thank you for your leadership on this topic. To share your experience it could be really useful for other NBS programs.

After reading your manuscript several times, I consider that the comments I can make are minor. The manuscript is very well explained, written and includes all the relevant aspects that should be reflected in a scientific article.

 Some minor comments are following:

Point 1: The introduction lacks some detail in terms of background of NBS program in Tennessee. It is suggested that the introduction is expanded to include a few lines about:

-         - How many diseases are screened

-          -How many NBS laboratories are there.

-          -From how many birth hospitals are samples received

Point 2: Please, add the descriptions of the abbreviations in Figure 1 for “TN, NBS, APHL, IT”.

Point 3: In Table 1: Why you don’t include “parents refused screening” in DBS screening rate indicator? This information could be important to know how many children are not screened and if you have an increase in rejections. I understand that, unless parents refused screening, all babies born in a birth hospital are screened.

Point 4: In Table 1: For unsatisfactory Specimen Rate indicator is recommended add and specify, in addition to not including items (as “DBS colleted less than 24h, etc…”), that it does include: for example, insufficient sample, bad quality sample, etc.

Author Response

Dear Reviewer,

Thank you for your comments! Please find below a point-by-point response to the comments made in your review.

Point 1

A line (line 28-29) was added to the first paragraph of the introduction describing this information for Tennessee Newborn Screening.

Point 2

The mentioned abbreviations have been spelled out in the caption for Figure 1.

Point 3

A footnote was added to Table 1 explaining that these specimens are not included when calculating DBS Screening Rate so as to not hold those specimens against the hospitals' performance when they have no control over them. The footnote also mentions that the number of those who refused is included in the form of a tooltip.

Point 4

Another footnote was added to Figure 1 that specifies what an unsatisfactory DBS is and states that they include DBS that are of poor quality, have incorrect information when submitted, or that have been mishandled. The footnote also mentions that the exact number for each of these types of unsatisfactory specimens is available in the form of a tooltip. 

Thank you! 

Reviewer 2 Report

This article is very interesting because Newborn Screening is one of the most important public health programs worldwide and a continuous improvement of quality parameters is essential for success.

Introduction

In my opinion, the introduction can be shortened, as some lines are redundant. For example, I would suggest omitting the sentence in lines 45-47 (same information in lines 40-41) and 48-51 (same information in lines 56-58). On the other hand, more information about the goals set in the NBS program and NewSTEPs in the USA would be helpful (nationwide, mandatory?….) and should be included

Materials and Methods

The important points discussed later, such as departmental and jurisdictional regulations, server restrictions, predefined schedules, state procedures and contracts should be mentioned in the first sentence

In my opinion, lines 89-97 are too detailed and too focused on the authors. I would recommend including this information in lines 88-89: To build an effective dashboard, experience with data visualization techniques and software, as well as profound knowledge of NBS laboratory and follow-up program data and processes, is highly desirable

Information in lines 83/84 is the same as in lines 103/104

Line 105-106: More information is needed on the reports studied to identify key metrics (literature, process what factors are included) and the source of the data tables used (i.e. where recorded and linked)

Tableau (line 102) and SAS (line 100) should be properly cited

Results

A sentence about the results of the survey should be included in this chapter

It would be nice to mention the URL to the dashboard and list the literature in Table1 for the state goals

Discussion

 Lines 201-212 are quit detailed

The experience with other dashboards should be discussed (i.e. Weiss et al 2018 BORN Ontario, other papers??), Wyoming uses a dashboard for NBS, NewSTEPS encourages and have dashboards…

The indicators differ from the NewSTEPs recommendations, selection for the Tennessee dashboard should be discussed, as it might be important for others (are the NewSTEP indicators not available, not important for the facilities….)

Literature

Website citation is inappropriate because URL and date of retrieval are missing; use of italics is not always identical

Author Response

Dear Reviewer,

Thank you for your comments! Please find below a point-by-point response to the comments made in your review.

Introduction

The lines 45-47 were rewritten along with the lines 43-44 to shorten the introduction and to specify that data dashboards are a specific tool of data visualization that are well suited for visualizing complex datasets.

Lines 48-51 were removed.

Added lines 54-56 to specify the reasoning behind the goals set for NBS (to ensure results are reported by day of life 7) and to state that they are set across the USA with the expectation that they will be strictly adhered to, but they are not mandatory.

Material and Methods

Added the points specified regarding regulations, restrictions, and procedures to lines 75-79.

Added the recommended line to lines 88-89 and rewrote lines 89-97 to be less detailed. The lines 95-99 now describe the dashboard workgroup as being composed of a fellow from the Association of Public Health Laboratories, TN NBS leadership, and a epidemiologist and nurse educator.

Omitted lines 103/104 and rephrased original lines 83/84, now 87-89, to include information relevant from lines 103/104. 

Lines 105-106 were expanded, now lines 106-110, to include how metrics were chosen (cited the reports being mentioned & explained targetable metrics that gave an overview of  NBS performance were selected).

Added the relevant information to further describe the tools Tableau and SAS (who produced them & where are they located), lines 101 and 103.

Results

Added lines 161-163 to the start of the results that contain a citation to the URL of the NBS dashboard.

Added the citation to the literature for the state-set goals, line 170.

Added lines 193-196 to the Results section on the Individual Indicator View describing how the survey influenced the design of this view.

Discussion

Cut much of the detail from lines 201-212, now lines 217-224, and kept only the main details needed to describe the impacts the dashboard has had for TN NBS so far.

Added lines 233-236 discussing how the review of existing dashboards contributed to the success of developing the TN NBS dashboard. Added these relevant details, lines 117-121, to the Materials and Methods section as well in the Dashboard Planning section.

Added lines 269-275 discussing why the indicators differ from the NewSTEPs indicators (patient privacy concerns and leadership decisions).

Literature

Corrected the references to ensure that URL and date of retrieval were included in website citations and that use of italics was standardized to book/journal titles.

Thank you!